# Exploring the Influence of News Consumption on Non-Muslim Australians' Attitudes towards Muslims

Jacqui Ewart [1,2,*] and Shannon Walding [2]

1   School of Humanities, Languages and Social Science, Griffith University, Nathan, QLD 4111, Australia
2   Griffith Criminology Institute, Griffith University, Nathan, QLD 4111, Australia
*   Correspondence: j.ewart@griffith.edu.au

**Abstract:** Research into news media representations of Muslims and their faith has focused mainly on how Muslims are portrayed in various types of news media and how stories about or involving them are framed. However, there has been very little attention paid to the effects of news consumption on attitudes towards Muslims. Accordingly, we wanted to explore a range of issues associated with news consumption levels and attitudes towards Muslims in Australia. The three objectives of this article are to: explore whether the amount of news consumed by respondents to an Australian survey influences the level of animosity they hold towards Muslims; determine how political viewpoint and religiosity influence the relationship between news consumption and animosity towards Muslims; and see whether engagement with Muslims influences the relationship between news consumption and animosity towards Muslims. Through a 2018 nationally representative sample of Australians, we target these objectives by investigating whether the amount of news that non-Muslim survey participants consume in a week influences the levels of anger they feel towards Muslims and how their self-defined religiosity, political viewpoint, and engagement with Muslims affect that relationship, while controlling for known drivers of anti-Muslim sentiment, such as demographic characteristics and knowledge about Muslims. We set our study in the contemporary context of mostly lab-based research that helps us understand how news media consumption affects particular types of people and whether there are commonalities in like-groups' responses to different types of news consumption; in this case, stories about Muslims and their faith. The findings of our research will be of interest to news media organizations and journalists wanting to know about the effects of their coverage of stories about Muslims and their faith and those wanting to improve that reportage. The results will also interest groups working on social cohesion efforts, those trying to improve inter-faith and inter-cultural relations, and academics investigating news media coverage of Muslims and Islam. Significantly, we find quantity of news consumption to lack effect on anger levels.

**Keywords:** Australian Muslims; Australian non-Muslims; attitudes to Muslims; news consumption

## 1. Introduction

Researchers have focused a great deal of attention on how Western mainstream news media covers stories about Muslims and their faith. Much of this coverage is negative, with a range of problematic practices associated with stories about or involving Muslims (Poole 2002; Saeed 2007; el-Aswad 2013). There is also a wealth of research that provides insights into the factors, mostly demographic, that characterize non-Muslims' attitudes, both positive and negative, towards Muslims in Europe, the United Kingdom, North America, Australia, and New Zealand (Erdenir 2010; Triandafyllidou 2015; Shaver et al. 2016; Shaver et al. 2017; Sibley et al. 2020; Dunn 2005; Walding and Ewart 2022). We know a lot less about whether news media consumption is a factor in the type of attitude non-Muslims hold towards Muslims, and how that works.

This article is set within the context of rising anti-Muslim and anti-Islam sentiment internationally (Erdenir 2010; Triandafyllidou 2015) and in Australia (Briskman and Latham

2017; Kolig 2006, 2010; Dunn 2005). There is also some research, albeit in its infancy, showing that news media consumption has an influence on attitudes towards Muslims (Shaver et al. 2017). While research already explored non-Muslim Australians' attitudes towards Muslims (Walding and Ewart 2022; Mansouri and Vergani 2018), we decided to explore whether non-Muslim Australians' consumption of news media affected the extent of anger they felt towards Muslims. We frame our investigation using media effects theory because research suggested that news media has a significant role in shaping attitudes of non-Muslims towards Muslims, particularly those who have little or no contact with Muslims. While research recognizes that social media plays a role in influencing the attitudes of non-Muslims towards Muslims, we focus on traditional news media rather than social media because we know from previous research (O'Donnell et al. 2021) that the majority of Australians still use news media as their primary source of information about Muslims and their faith. News consumption's influence on attitudes towards Muslims is an area that has received surprisingly little attention in Australia. Our study draws data from a representative 2018 National Social Survey of 1017 Australians who did not self-identify as Muslims.

This article was driven by a number of insights provided by the research. They included the significant body of research into problematic news coverage of Muslims and their faith, in Australia and internationally, along with the literature that largely highlights the negative effects of news media coverage of non-Muslims' attitudes towards Muslims. Shaver et al. (2017) and colleagues' study of news consumption and anger amongst New Zealanders contributed to our desire to see whether increased news consumption had a similar connection to increased anger towards Muslims in Australia, along with whether that relationship was consistent across the political spectrum. Our finding that the effect of political viewpoint on attitudes towards Muslims was inextricable from the effect of how important religion was to a participant meant that we checked for the consistency of the relationship across those two concepts.

As such, we set out to examine the following three hypotheses:

**H1.** *Media effects: How are news consumption and animosity towards Muslims related? If media effects influence animosity towards Muslims, then increases in hours of news consumption will be related to higher levels of anger, over and above the effects of other factors.*

**H2.** *Confirmatory bias: How does the combination of political viewpoint and religiosity influence the relationship between news consumption and animosity towards Muslims? If confirmatory bias exists, political viewpoint should neutralize media effects on anger.*

**H3.** *Contact with Muslims: Media effects theory rests on media portrayal of Muslims in the absence of personal knowledge through engagement with Muslims. How does engagement with Muslims influence the relationship between news consumption and animosity towards Muslims? If media effects influence animosity, the relationship between news consumption and anger should be stronger among those with less/no personal engagement with Muslims.*

We found no evidence that news consumption levels have an effect on the amount of anger non-Muslim Australians hold towards Muslims. However, we found that a range of other factors, including religiosity, politics, knowledge about Muslims, and personal engagement with Muslims have an influence on the level of anger survey participants hold towards Muslims, although effects were small. Political viewpoints and religiosity together have a notable role to play in anger levels towards Muslims. However, the lack of relationship between news consumption levels and anger towards Muslims does not change across the political/religiosity spectrum. We discuss the implications of our findings and suggest related areas for further and future research.

## 2. Literature

There is a wealth of literature about the ways in which stories about or involving Muslims are covered in Western news media (Poole 2002; Karim 2006; Poole and Richardson

2006; Zelizer and Allan 2011). This is a well-worn terrain that continues to be a source of interest to researchers. Despite their relatively small presence in Australia, at around 2.6 percent of the population (Australian Bureau of Statistics 2017), and their long history of association with and presence in Australia both pre and post colonization (Deen 2009; Ganter 2008; Saeed 2003), Muslims attract a significant amount of negative news media coverage in Australia (Ewart and O'Donnell 2018).

There are high levels of anti-Muslim sentiment in Europe, America, and Australia. These attitudes are characterized by varying degrees of anger, fear, and concern about Muslims being a threat to the safety of non-Muslims (Erdenir 2006; Field 2007; Brockett et al. 2009; Erdenir 2010; Cheng 2015; Triandafyllidou 2015). The research sheds light on the demographic characteristics of those who hold varying degrees of anger and warmth towards Muslims, although findings are often contradictory.

Research highlights the problematic nature of much news media coverage of stories about Muslims and Islam in some Australian news media (Ewart and O'Donnell 2018). However, there is far less research, both in Australia and elsewhere, that explores the effects that the consumption of news media has on attitudes towards Muslims. The research undertaken in this area is often by those in the psychology field, and it is here that media effects theory seems to attain significant traction in the framing and findings of these studies (Williamson 2019; Crawford 2014; Doosje et al. 2009; Unnever and Cullen 2010).

**3. News Media Coverage of Muslims**

A study of 300 academic articles about news media coverage of Muslims and the construction of their identities identified that much of this research focused on news media in the United States of America, the United Kingdom, and Australia (Ahmed and Matthes 2017). There is evidence that some mainstream news media coverage was a problem pre-9/11 as Zelizer and Allan (2011) and others argue (Richardson 2001; Manning 2003; Nacos and Torres-Reyna 2007). However, post-9/11 researchers analyzing news coverage suggest a widespread change in news media engagement with, and reportage about, Muslims (Poole and Richardson 2006). This research identifies a range of problems present in stories about Muslims, from stereotypes to the publication of misinformation, and the conflation of Islam with terrorism. Researchers highlight that a lot of traditional news media coverage focuses on negative stories, conflating the religion with violence (d'Haenens and Bink 2006; Ahmed and Matthes 2017). Such coverage is identified as contributing to community unease, tensions, feelings of discrimination, victimization, isolation, and social division (Grossman and Tahiri 2013). The lack of Muslim voices in stories about Muslims and their religion is also problematic (d'Haenens and Bink 2006; Richardson 2001; Karim 2006; Moore et al. 2008).

Brown et al. (2015) examined the responses of Muslims to news media coverage of stories about them, finding participants thought stories generally took the following themes: conflating Muslims with terrorism; portraying Muslim countries as strict and conservative; and positioning Muslims and their countries as dirty, backward, and uneducated. In 2010 Ülkü Güney looked at how young Asian people in Bradford, a socio-economically impoverished mill town in Britain, perceived news coverage of various wars in the Middle East. Güney (2010) found these young people experienced negative reactions and felt that reportage strengthened non-Muslims' perceptions of the connections between Muslims and terrorism.

Anne Aly (2007) studied the reactions of Australian Muslims to news coverage about them and their religion. She found that many of her study participants intensely distrusted news media, and that Australian Muslims viewed news media discourse as anti-Muslim even when other explanations for approaches to news reporting were available. Another Australian study by Murphy et al. (2015) examined the responses of some Muslims to news media coverage of stories about Islam, finding they experienced increased feelings of stigmatization as a result of the coverage's negative descriptions of Australia's Muslim

communities. In addition, their study found participants considered news coverage often comingled Islam with terrorism.

## 4. Attitudes towards Muslims

The research into attitudes towards Muslims is important in the context of exploring, as we are, whether news media consumption is one of the identified drivers of anti or pro-Muslim sentiment.

Research into the drivers of anti-Muslim sentiment focused on the two schools of thought that such sentiments are driven by either Muslimophobia or Islamophobia (Erdenir 2010; Triandafyllidou 2015). Muslimophobia is the dislike of the Muslim people; as Burak Erdenir (2010, p. 28) explains, it 'targets Muslims as citizens or residents of European countries rather than Islam as a religion'. This is different from Islamophobia, criticized as a vague concept by some researchers, which Erdenir explains is focused on religious discrimination. Anti-Muslim sentiments in Europe focus on perceptions that it is Muslims' 'culture, lifestyle, and values' (Erdenir 2010, p. 37) that are the problem, rather than the religion per se. While in European politicians and news media focus on Muslims (Erdenir 2010), in Australia, it is the religion of Islam that attracts more negativity from non-Muslims than Muslims themselves (Ewart et al. 2021).

As Brockett et al. (2009, p. 241) pointed out, when approaching the question of attitudes towards Muslims, many scholars do so by attempting to 'operationalize an underlying attitude of prejudice, fear or loathing linked to concepts such as racism and Islamophobia'. Some researchers identified that characteristics, such as gender, education levels, religiosity, and political leanings have a significant influence on levels of anger and negative attitudes non-Muslims have towards Muslims (Helbling and Traunmuller 2020; Hellevik 2020; Gusciute et al. 2021). However, there is no one common set of demographic characteristics that are found to belong to those who hold anti-Muslim sentiments, and conversely, those that hold warmth towards them.

In addition, Muslims and immigrants are often conflated in surveys of attitudes towards Muslims. Bakker-Simonsen and Bonikowski (2020, p. 114) highlight the crucial differences in the attitudes that non-Muslims hold towards immigrants and Muslims. Drawing on research data from 41 countries in Europe, they found that 'varieties of national self-understanding are predictive of anti-Muslim attitudes, above and beyond dispositions toward immigrants'. More evidence of the conflation of Muslims with immigrants, and the subsequent effects on the attitudes that non-Muslims hold towards Muslims, was highlighted by a study undertaken in Germany, a country that has a significant number of Muslim residents. Wallrich et al. (2020) drew on data from a 2016 survey of Germans that focused on foreign groups residing in Germany. That survey (Wallrich et al. 2020, p. 2195) revealed that when respondents are asked about foreigners, they 'disproportionately think of groups who are Muslim, and that such salience is associated with more negative attitudes towards "foreigners"'. Wallrich and colleagues found their study participants held more antipathy towards Muslims than towards other 'foreigners'.

On the other hand, Helbling and Traunmuller (2020, p. 811) revealed that anti-Muslim sentiment amongst study participants in the United Kingdom was primarily defined by their perception of Muslims' religious behavior. This was more significant in citizens' uneasiness with Muslims than ethnic or religious identity. One important finding from their survey (Helbling and Traunmuller 2020, p. 822) was that 'liberals—who place less emphasis on national security—have more negative feelings about religious radicals than conservatives do'. Study participants also believed that the religion of Islam and Muslims' religious behaviors were at odds with the values of the UK, such as democracy and liberal values. However, in Norway, two Poynting studies by Ottar Hellevik (2020, p. 120) undertaken in 2011 and 2017 point to a 'positive correlation between the share of immigrants in a local community and positive attitudes towards them', suggesting that familiarity with Muslim people ameliorated feelings of uneasiness about their behavior or identity. Nevertheless, their surveys highlighted that men, older people, and those with lower

education levels expressed the highest levels of Islamophobia. Elsewhere in Europe, Gusciute et al. (2021) analyzed data from the 2014 European Social Survey, involving 26,000 participants, revealing that women, and in particular conservative women, held higher levels of anti-Muslim sentiment than men. They also found a higher level of opposition to Muslim immigration in the countries involved in the survey than other types of immigration.

In New Zealand, similarly to Australia, Muslims proportionately make up only a very small percentage of the country's population at one percent (StatsNZ 2014). However, they have attracted a great deal of research attention. Greaves et al. (2020, p. 260) drew on data from a survey undertaken in 2018/2019 to reveal that 'older people, New Zealand Europeans, men and those with more right-wing attitudes' were amongst those with increased levels of negativity towards Muslims, while these groups also perceived that Muslims posed a high level of threat. A third of survey participants saw Muslims as a threat, with a higher threat perception level than the participants had of any other religious group. However, a third of those surveyed thought Muslims posed no threat. Another New Zealand study worthy of mention was undertaken by Shaver et al. (2016) and focused on levels of warmth New Zealanders held towards Muslims. They found that non-Muslim New Zealanders who attended churches or other places of worship regularly held greater warmth towards Muslims than those who did not. Shaver and colleagues posited that uneasiness with faith might explain the reduced levels of warmth amongst non-religious survey participants. One fascinating association to emerge in their study was the connection between Muslims' ethnic background and higher levels of anti-Muslim prejudice, with study participants holding lower levels of warmth towards Arab Muslim immigrants than other immigrants. The demographic characteristics of survey respondents who hold lower levels of warmth towards Muslims and Arabs included men, the politically conservative, those with low education levels, and the unemployed.

In Australia, some politicians gained popularity on a platform that promotes fear of Muslims and positions them as outsiders who do not possess or subscribe to Australian values (Reid 2019; Forrest et al. 2020). The One Nation political party, with its anti-Islam platform, added significantly to the increase in anti-Muslim sentiment in Australia (Poynting 2004), while Muslims are presented by some political parties (Aly 2007) and extreme groups (Miller 2017) as a political and cultural threat. While relatively few studies evaluated non-Muslim Australians' attitudes towards Muslims, extant research highlighted that non-Muslims consider that some of the practices of Islam do not fit within Western society and there is an uneasiness around perceptions of Muslims' attitudes towards women and children (Sniderman and Hagendoorn 2007; Adida et al. 2016; Bail 2016). Research by Kristy Campion (2019) and Poynting and Briskman (2018) highlights that politicians and commentators have a key role in normalizing intolerance towards Muslims in Australia.

Islamophobia in Australia was the subject of a study by Dunn et al. (2021) in which two surveys administered in 2015 and 2016 were used to identify the typologies of groups with varying perceptions of Islam. Of those surveyed, 87 percent had some concerns about Muslims, with 13 percent being identified as Islamophobes, 24 percent recording that they had some concerns about Muslims but were unsure about diversity in the population, and 50 percent having progressive beliefs about diversity, but some concerns about Muslims. Meanwhile, 13 percent identified as progressives with no concerns about Muslims. While many of the demographic characteristics of those surveyed were discarded from their analysis, Dunn and colleagues did highlight that older Australians were more likely to be Islamophobic. Another study by Mansouri and Vergani (2018) looked at attitudes held by non-Muslim Australians towards Muslims by measuring prejudice through the operationalization of the desire participants had to be socially distant from Muslims. They found that those who had more contact with Muslims knew more about Islam and held less prejudice towards Muslims. Mansouri and Vergani also identified that being more conservative politically and less well educated had a significant influence on prejudice levels towards Muslims, but gender and religiosity did not have a similar effect.

Colic-Peisker et al. (2020) surveyed 1020 non-Muslims in the Australian cities of Sydney and Melbourne in 2019. Their study participants, who were Christians, identified as being more Islamophobic than those who had no religion or identified with other non-Islamic religions. Those surveyed in the 18–24 age group and who had attended university or completed years 11 or 12 were also identified as having low levels of Islamophobia. As many other surveys show, those in Colic-Peisker and colleagues' study who had a lower socio-economic status, had higher levels of Islamophobia.

## 5. News Media Consumption and Attitudes towards Muslims

Whether and to what extent news consumption might influence attitudes towards Muslims is deeply contested terrain (Neuman and Guggenheim 2011; Scheufele and Tewksbury 2007). Many of those who undertake research in this area focus on media effects theory, which emerged from studies of mass communication in the 1920s and gained a stronger foothold in the 1950s as communication researchers began to focus on television (Valkenburg et al. 2016, p. 317). Media effects research is broad in its scope, encompassing several academic fields and a wide range of topics; for example, the effect of 'exposure to media violence on aggression and of advertising on purchase behavior, to the effects of Internet use on political engagement and of Facebook use on loneliness' (Valkenburg et al. 2016, p. 317). Much of the research focused on enumerating the effects of media on behavior through psychologists measuring the effect of specific media consumption on self-reported responses.

While effects were certainly observed and recorded, there is disagreement about the extent of the influence that news media has on attitudes and behaviors with some, such as Valkenburg et al. (2016, p. 317) and colleagues, arguing that the effect of media consumption on behaviors of 'large heterogeneous groups' is limited. Valkenburg and colleagues' work theorized that there were five main features of media effects theory: media users are selective in their media use; the properties of media consumed are predictors, along with the types of media influencing their effects; and that media effects are indirect, conditional, and transactional. However, other researchers, for example Miles-Novelo and Anderson (2020), posit that news media consumption has considerable influence on those exposed to it, with attendant changes in behaviors and attitudes. Slater (2007, p. 290) undertook a review of key literature about media effects, including a discussion that is pertinent to our article that looked at the 'effects of reinforcing spirals on individual media consumers'. Slater (2007, p. 290) found the literature highlighted that there was a 'mutually reinforcing' process at work with media selection having an effect on users' 'social identity and attitudes as well as the behaviors of group members'.

The complexity of media effects theory (Scheufele and Tewksbury 2007) is demonstrated in part by the raft of studies that academics from the psychology discipline have undertaken that examine how audience members receive and respond to news media content. The research is polarized, with one school of thought suggesting news media has little to no effect when it comes to changing the opinions or views of those consuming it, and the other firmly supporting the opposite view (Neuman and Guggenheim 2011).

McCombs and Reynolds' (2002) work is useful for finding a way through the polarized landscape of media effects theory. They suggest that news media 'plays a key role in the construction of our pictures of reality' (McCombs and Reynolds 2002, p. 2). If we take this as a starting point, then theories such as news framing, agenda setting and priming all help us to consider '(a) how news messages are created, (b) how they are processed, and (c) how the effects are produced' (Scheufele and Tewksbury 2007). Another important part of the process of the construction of news stories is the decisions that journalists make that 'significantly influence their audience's picture of the world' (McCombs and Reynolds 2002, pp. 5–6).

Research that examined media effects, particularly in relation to news consumption and its effects on non-Muslims' attitudes towards Muslims, reveals that news media consumption has *some* level of influence on attitudes and behaviors, although that influence

varies. For example, Miles-Novelo and Anderson (2020, p. 59) examined the 'effects of exposure to media stereotypes of Muslims and the psychological processes underlying them' amongst American news consumers. The (Miles-Novelo and Anderson 2020, p. 59) approach they took was situated in media effects theory, with these researchers suggesting that 'we may base most of our beliefs and attitudes about specific groups of people not from real-world experience with members of that group, but from what we see in the media'. They propose that (Miles-Novelo and Anderson 2020, pp. 59–60) 'most negative attitudes towards Muslims stem from exposure to negative media portrayals'. Furthermore, they wrote that those who have limited or no interactions with specific outgroups are likely to cultivate very negative attitudes towards the outgroups. Miles-Novelo and Anderson argue that Americans' experiences with Muslims are limited and that, because they have little direct contact with Muslims, their negative attitudes towards Muslims are influenced by media stereotypes of Muslims.

Another study of note and interest to news media influence on attitudes was undertaken by Saleem et al. (2017). They examined how news media stereotyping of Muslims affected the support non-Muslim Americans gave to public policies that were specifically and exclusively harmful to Muslims. There was a relationship between their study participants' political conservatism and shifts in support for policies that harmed Muslims, highlighted by participants being shown examples that countered stereotypical news coverage. When shown these examples, participants became more unfavorable towards the aforementioned harmful policies.

Many researchers found links between the high levels of anxiety news consumers experience when they read stories about terrorism and their attitudes towards Muslims. For example, Haner et al. (2019) found that women and those more religiously inclined, along with people who favored conservative politics, had concerns about terrorism, and thus were more likely to support policies that were anti-Muslim. In examining how news media influenced the perspectives of their study participants in relation to terrorism threats, they (Haner et al. 2019, p. 12) found participants' interest in political news was:

> negatively associated with both fear and worry; this finding was counter to our expectations. Although, as noted, our measure is not specific to exposure to terrorism-related news, this finding suggests that the news media in general may not have consistent fear-arousing effects on Americans. Rather, it is possible that greater interest in political news indicates more awareness of current issues and a decreased likelihood of emotional responses to terrorism.

Perception of news bias was the focus of a Swedish study that surveyed just under 1000 people. That study set out to determine participants' perceptions of news bias when it came to coverage of Muslims and their faith in Swedish news media. Dorothea Arlt (2021) explored study participants' voting intentions on an initiative involving a ban on veiling amongst Muslim women and how they were influenced by news coverage. The most relevant influencing factors that influenced their voting intentions were their attitudes towards Muslims and their faith, the participants' political leaning, and whether they had personal contact with Muslims. Arlt found in contrast (Arlt 2021, p. 9) 'exposure to political information via traditional news media and social media was not associated with bias perceptions'. Interestingly, study participants (Arlt 2021, p. 9) who thought that the news media minimized problems in the reporting of stories about Muslims and their faith in Sweden intended to vote for the 'national ban on wearing burkas or niqabs in public'.

In a New Zealand study, Shaver et al. (2017) set out to explore whether there was a connection between news exposure and anti-Muslim sentiment in New Zealand through a national survey administered in 2013. They found (Shaver et al. 2017, p. 1) 'greater news exposure is associated with both increased anger and reduced warmth toward Muslims'. The anti-Muslim sentiment they identified in their survey participants existed across the range of political affiliations, although they did note that news media coverage in 2013 involved more of a focus on Muslim extremism, as terrorism attacks increased significantly the previous year. The authors (Shaver et al. 2017, p. 1) make some more general points

about the body of news media consumption literature, pointing out that 'people tend to interpret the news in support of pre-existing beliefs' in looking for coverage that ratifies their opinions, while dismissing coverage that does not endorse their pre-existing beliefs. Their findings (Shaver et al. 2017, p. 1) that 'greater news exposure is associated with both increased anger and reduced warmth toward Muslims' were particularly interesting because New Zealand is considered a religiously and culturally tolerant country with no history (at the time) of regional Muslim conflicts.

In Australia, where Muslims were the focus of wedge politics (Manning 2004; Dunn and Kamp 2016), researchers began to explore the question of the sources Australians used to inform themselves about Muslims almost two decades ago. Kevin Dunn (2005) found that Australians know relatively little about Muslims and their religion. Using data from a 2003 survey of Australians, Dunn (2005) explored the study participants' knowledge of Muslims and their faith. He found that for some of the participants, the way they thought about and differentiated Muslims in the Middle East from those in Indonesia was shaped by the news media. Those survey respondents (Dunn 2005, p. 27) who had direct contact with Muslims tended to have more positive perceptions of Islam that 'run counter to the dominant Western media image of Islam'.

A small, not representative, study involving Queenslanders by Rane (2010), revealed that, of those surveyed, 80 percent relied on news media as their main source of information about Muslims and their faith. Following this, there was a lull in research that focused on Australians' attitudes towards Muslims and the sources they used to gain information about Muslims and their faith. In the early 2020s another study emerged that had significance for this article. O'Donnell et al. (2021) explored the main sources of information for non-Muslim Australians about Muslims, finding that 80 percent of journalists they surveyed thought mainstream news media was the primary source of information for non-Muslim Australians abut Muslims. In comparison, that study found that just under half of their non-journalist survey participants used mainstream news media as their main source of information about Islam and Muslims (49.27 percent and 45.89 percent, respectively). They identified (O'Donnell et al. 2021, p. 1031) that 'Muslim people, books, social media and family and friends are also important primary sources of information'.

There are three key aspects of the extant research that informed our approach. They are: the news media's role in influencing attitudes towards Muslims; the influence a person's political leaning has on their attitudes towards Muslims; and the role of contact with Muslims on attitudes towards them. We also know from the research that, in Australia, news media is a key source for information about Muslims and an influencer in attitudes towards them, but for non-Muslims, contact with Muslims counters negative news media portrayals. Building on this literature, we explore whether media effects theory is supported in non-Muslim Australians' attitudes towards Muslims in as much as the amount of consumption of news affects levels of anger. We base our study to some extent on that of Shaver et al. (2017) and colleagues' because it is the only study that we were able to locate that looked specifically at how levels of news consumption affected attitudes towards Muslims, and the prima facie similarity between Australia and New Zealand as societies.

## 6. Method

### 6.1. Plan of Analysis

Shaver et al. (2017) found that greater news consumption among New Zealanders was associated with higher levels of anger towards Muslims, over and above anger towards other outgroups, and over and above other contributory factors. The authors also found that relationship to be consistent across the political spectrum. In this study, we aim to discover whether similar results might be obtained using an Australian sample, with similar control variables and the inclusion of some additional known drivers of anti-Muslim sentiment. Our exploration of empirical support for elements of media effects theory in the Australian context of anti-Muslim sentiment centers on the following foci:

1.  H1 Media effects: How are news consumption and animosity towards Muslims related? If media effects influence animosity towards Muslims, then increases in hours of news consumption will be related to higher levels of anger, over and above the effects of other factors.

2.  H2 Confirmatory bias: How do the combination of political viewpoint and religiosity influence the relationship between news consumption and animosity towards Muslims? If confirmatory bias exists, political viewpoint should neutralize media effects on anger.

3.  H3 Contact with Muslims: Media effects theory rests on media portrayal of Muslims in the absence of personal knowledge through engagement with Muslims. How does engagement with Muslims influence the relationship between news consumption and animosity towards Muslims? If media effects influence animosity, the relationship between news consumption and anger should be stronger among those with less/no personal engagement with Muslims.

The scope of our study requires a more traditional approach to data analysis than that undertaken by Shaver et al. (2017); however, we do not aim to replicate their study, but instead to examine whether Australian survey data supports similar elements of media effects theory. As such, we undertake a series of Tobit/censored regression models explaining anger towards Muslims, including our key variables of interest (news consumption, political viewpoint and religiosity, and personal acquaintance with Muslims) and other known drivers of anti-Muslim sentiment (knowledge about Muslims and selected demographic characteristics).

Age, gender, education, socio-economic status, and residence in urban or regional areas are inextricably linked in Australian samples (Walding and Ewart 2022), and as discussed, the variability of effects on attitudes towards Muslims according to contexts of characteristics rather than each characteristic alone resulted in an inconsistency in evidence of relationships between demographics and Muslimophobia in samples from a range of countries. Rather than engaging in detail with the modelling difficulties of such interrelated concepts, we examine the effect of news consumption on Muslimophobia, and the mediating effects of political viewpoint and religiosity, as well as personal engagement with Muslim people, on a representative sample of Australians while controlling for the effect of their demographic characteristics.

All relationships are modelled using Tobit regression analysis with robust standard errors in STATA 16.1. Tobit regression is preferred due to the censored nature of the dependent variable, anger towards Muslims, which is limited to scores between 1 and 7, with clustering at the boundary value of 1 (participants reporting no anger). This suggests that the assumption of an underlying latent variable represented by the observed scale of anger towards Muslims is appropriate in this case (as proposed by Tobin 1958). Checks of Akaike and Bayes information criteria (AIC and BIC; smaller values indicate better fit: see Long and Freese 2014) strongly recommended the Tobit specification over ordinary least squares. Because we are testing theory, results are reported with an emphasis on the presence of effect and how well a theoretical model fits the data. Model fit was assessed using likelihood ratio tests for overall model significance and multiple pseudo-R squared measures, of which we report the Cragg–Uhler/Nagelkerke calculation, and preferred a model determined through comparison of the AIC and BIC.

*6.2. Data*

To obtain a nationally representative sample of Australians with which to test the media effects and confirmatory bias hypotheses in the Australian context, we commissioned questions measuring our concepts of interest through the Life in Australia[TM] probability-based online panel. This panel is unique in Australia because of its improved representativeness through utilizing either telephone or online data collection depending on the participant's preference. In the September 2018 wave, 1400 of the 2800-strong panel of

Australian adults were invited to participate, with a completion rate of 78.3 percent (see Kaczmirek et al. 2019 for further details about the recruitment of the panel).

The Social Research Centre (a subsidiary of Australian National University) administered the survey to a sample of 1096 English-speaking residents of Australia aged over 18 years in September 2018. Questions were supplied by researchers on a cost-shared basis, along with a selection of core demographic and health questions. As participants were asked to identify their religion and this study's focus is on the effects of media consumption on non-Muslim people, those who identified as Muslim were removed from our sample for analysis, as were all participants with missing data on any variables for analysis. The resulting sample consisted of 1017 Australian adults with ages ranging from 18 to 94 years (M = 53.39, SD = 16.99). Almost 60 percent of the sample was part of the workforce at the time of the survey, while 44.94 percent attended university, and another 27.93 percent attended technical institutions. The sample was mostly urban (65.78 percent), female (51.13 percent), and less disadvantaged, with 68.44 percent coming from areas with middle to high scores on the index of relative socio-economic disadvantage (IRSD).

Among other questions, participants were asked about their regular engagement with Muslim people and anger towards them, their political viewpoint and the importance of religion in their lives, as well as the amount of news they consumed in a week. In addition, participants answered questions testing their knowledge about Muslims and Islam in general (see Appendix A).

*6.3. Measures*

Participants were asked about their level of anger towards Muslim people on a scale from 1 to 7, with 37.36 percent reporting no anger (1), 34.02 percent claiming neutral feelings (4), and 1.97 percent rating their anger at the maximum on the scale (7). In order to account for both the known effects of having personal engagement with Muslims on Muslimophobia, and the alternate source of knowledge about Muslims that provides, participants nominated how often they were in the company of one or more Muslim people, with almost six out of 10 indicating never or less than monthly. For this group, it is likely that the media are their main source of information about Muslims (O'Donnell et al. 2021).

Key to examining the media effects theory, survey participants estimated the number of hours in the previous week that they watched, listened to, or read the news. These estimates ranged from zero to 45 h, with 85.64 percent nominating 10 h or less.

Almost 40 percent of the sample indicated they had no religion, while the remainder rated the importance of their religion or spiritual group to how they saw themselves on a scale from 1 (not important) to 7 (very important). By including those with no religion as scoring a zero on that scale, our sample, on average, rated the importance of religion to their view of themselves at 2.66 (SD = 2.67). Participants were also asked to place themselves on a political spectrum where zero was the extreme left and 10 the extreme right. The most common response was a 5 (38.35 percent; M = 4.82, SD = 2.02). When these variables were examined in conjunction with anger towards Muslims, it became clear that where religion was unimportant to a participant, their political viewpoint was the stronger driver of their attitude towards Muslims, whereas if their political viewpoint was neutral, the importance of religion was the stronger driver. As such, these two variables were inextricably linked when examining Muslimophobia.

Other known drivers of Muslimophobia examined in other studies were included here, including demographics such as age, gender, education, urban living, and participation in the workforce, as well as knowledge about Muslims. A series of questions designed to test basic knowledge about Muslims was asked, enabling participants to obtain a score out of 14. On average, participants scored 10.22 out of 14 (SD = 2.35).

**7. Results**

Results of five Tobit regression models are reported in Table 1, including measures of fit. Model 1 depicts the relationship between hours of news consumed in the previous

week and anger towards Muslims. There is no evidence to support that a model including hours of news consumed on its own is any more predictive of anger towards Muslims than a model without any explanatory variables (all measures of model fit and statistical significance support that conclusion). We then tested models including combinations of other known drivers of Muslimophobia, as well as the interaction between hours of news consumed and levels of personal engagement with Muslim people in order to further investigate the existence of evidence supporting the media effects hypothesis, as well as interactions between the hours of news consumed and the participant's religiosity and political viewpoint to examine evidence supporting the confirmatory bias hypothesis.

As such, Model 2 includes political viewpoint and religiosity as well as the interaction between the two, along with the frequency with which the participant reported spending time with Muslim people and their knowledge about Muslims. While the overall model increased in explanatory power and fit, and the effects of political viewpoint (b = 0.37, $p < 0.001$) and religiosity (b = 0.48, $p < 0.001$) along with their interaction (b = $-0.06$, $p < 0.001$), as well as the effects of time spent with Muslim people (using a post-estimation Wald test, $F(4, 1008) = 4.29$, $p = 0.002$) were statistically significant, the effect of hours of news consumed on anger towards Muslims remained consistent with the single variable model (b = 0.01, 95%CI [$-0.02$, 0.04]).

Inclusion of measures of demographic characteristics in Model 3 had no meaningful effect on either the explanatory power of the model or the size and strength of the relationship between hours of news consumed and anger towards Muslims, supporting our previous finding that the complexity of demographic characteristics' influence on Muslimophobia in Australia, and likely other jurisdictions, creates confusion in interpreting effects through the use of regression modelling (Walding and Ewart 2022). As a result, we chose not to include this block of variables in any further models in the interests of model parsimony. Model 4 included an interaction between hours of news consumed and the measures of religiosity and political viewpoint, while Model 5 instead included an interaction between the hours of news consumed and the frequency of personal engagement with Muslims. None of the models tested demonstrated any significant or substantive direct relationship between hours of news consumed and anger towards Muslims, leading us to conclude that we did not find evidence to support the media effects hypothesis (H1) in this context.

Model 4's interaction of hours of news consumed with religiosity and political viewpoint was intended to examine the confirmatory bias hypothesis. Evidence of changes in the effect of news consumption according to religiosity and/or political viewpoint would indicate that the news media has more effect on the consumer if it confirms a viewpoint already held. There was no evidence of the statistical significance of any such interaction in Model 4 (all interaction terms were approximately equal to zero, with none demonstrating statistical significance, while the coefficient of hours of news consumed lost precision in its confidence interval and Model 4 was not the preferred fit to the data through any measures examined), leading to the conclusion that the confirmatory bias hypothesis (H2) is not supported by our data.

In Model 5, we examined whether the effect of hours of news consumption differed according to the level of frequency of a participant's time spent with Muslim people. Again, the media effects hypothesis would be supported if we found that the relationship between news consumption and anger towards Muslims changed according to level of personal engagement with Muslims, because it is assumed that people who spend more time with Muslims are less likely to be affected by the news media because they obtain less of their information from the media. Again, we did not find evidence of this interaction between variables, or an improved model fit, and concluded that the relationship between news consumption and anger is not stronger among those with less/no personal engagement with Muslims (H3).



**Table 1.** Model comparison: Predicting anger towards Muslims using Australian survey data.

| | Model 1 | | Model 2 | | Model 3 | | Model 4 | | Model 5 | |
|---|---|---|---|---|---|---|---|---|---|---|
| | Estimate | 95% CI | Estimate | 95% CI | Estimate | 95% CI | Estimate | 95% CI | Estimate | 95% CI |
| **DV: Anger towards Muslims** | | | | | | | | | | |
| Hours of news consumed | 0.01 | [−0.03, 0.04] | 0.01 | [−0.02, 0.04] | 0.01 | [−0.02, 0.04] | 0.04 | [−0.07, 0.15] | −0.02 | [−0.09, 0.04] |
| Importance of religion | | | 0.37 *** | [0.19, 0.55] | 0.36 *** | [0.18, 0.53] | 0.38 ** | [0.14, 0.63] | 0.36 *** | [0.18, 0.54] |
| Political viewpoint | | | 0.48 *** | [0.34, 0.61] | 0.43 *** | [0.30, 0.57] | 0.53 *** | [0.34, 0.72] | 0.48 *** | [0.34, 0.61] |
| Importance of religion#Political viewpoint | | | −0.06 *** | [−0.10, −0.03] | −0.06 *** | [−0.09, −0.03] | −0.07 ** | [−0.12, −0.03] | −0.06 *** | [−0.10, −0.03] |
| Time spent in the company of one or more Muslim people | | | | | | | | | | |
| <Monthly vs. Never | | | −0.55 * | [−1.00, −0.09] | −0.49 * | [−0.96, −0.03] | −0.56 * | [−1.01, −0.10] | −0.75 * | [−1.40, −0.11] |
| Monthly vs. Never | | | −1.01 *** | [−1.56, −0.47] | −0.93 *** | [−1.47, −0.38] | −1.02 *** | [−1.56, −0.48] | −1.01 * | [−1.81, −0.21] |
| Weekly vs. Never | | | −0.68 * | [−1.22, −0.13] | −0.61 * | [−1.17, −0.05] | −0.68 * | [−1.23, −0.13] | −1.17 ** | [−1.93, −0.41] |
| Daily vs. Never | | | −1.03 ** | [−1.71, −0.36] | −0.88 * | [−1.59, −0.16] | −1.04 ** | [−1.72, −0.36] | −1.40 ** | [−2.30, −0.49] |
| Knowledge about Muslims score | | | −0.13 *** | [−0.21, −0.06] | −0.12 ** | [−0.19, −0.05] | −0.13 *** | [−0.21, −0.06] | −0.13 *** | [−0.21, −0.06] |
| Male vs. Female | | | | | 0.34 * | [0.01, 0.68] | | | | |
| Age (years) | | | | | 0.00 | [−0.01, 0.01] | | | | |
| Regional vs. Urban | | | | | 0.31 | [−0.04, 0.67] | | | | |
| In the workforce vs. not | | | | | −0.08 | [−0.49, 0.34] | | | | |
| Highest education level | | | | | | | | | | |
| Technical institution vs. < high school | | | | | 0.01 | [−0.53, 0.55] | | | | |
| High school graduate vs. < high school | | | | | −0.38 | [−1.04, 0.28] | | | | |
| University graduate vs. < high school | | | | | −0.60 * | [−1.12, −0.07] | | | | |
| Hours of news#Importance of religion | | | | | | | 0.00 | [−0.03, 0.02] | | |
| Hours of news#Political viewpoint | | | | | | | −0.01 | [−0.03, 0.01] | | |
| Hours of news#Importance of religion#Political viewpoint | | | | | | | 0.00 | [−0.00, 0.01] | | |
| Time spent in the company of one or more Muslim people#Hours of news | | | | | | | | | | |
| <Monthly vs. Never | | | | | | | | | 0.03 | [−0.05, 0.12] |
| Monthly vs. Never | | | | | | | | | 0.00 | [−0.10, 0.10] |
| Weekly vs. Never | | | | | | | | | 0.08 | [−0.01, 0.17] |
| Daily vs. Never | | | | | | | | | 0.06 | [−0.05, 0.17] |
| Constant | 2.28 *** | [2.01, 2.55] | 1.82 ** | [0.72, 2.93] | 1.94 ** | [0.52, 3.36] | 1.62 * | [0.31, 2.93] | 2.04 *** | [0.88, 3.20] |
| Var(e) | 7.29 *** | [6.44, 8.14] | 6.41 *** | [5.66, 7.16] | 6.25 *** | [5.52, 6.99] | 6.41 *** | [5.66, 7.15] | 6.38 *** | [5.63, 7.12] |
| Cragg-Uhler/Nagelkerke pseudo R-sq | 0.000 | | 0.112 | | 0.108 | | 0.114 | | **0.116** | |
| AIC | 3708.67 | | 3606.96 | | **3600.24** | | 3612.29 | | 3610.79 | |
| AIC/n | 3.65 | | 3.55 | | **3.54** | | 3.55 | | 3.55 | |
| BIC | 3723.44 | | **3661.13** | | 3688.89 | | 3681.23 | | 3684.65 | |
| Log lik. (Intercept only = −1851.40) | −1851.33 | | **−1792.48** | | −1782.12 | | −1792.14 | | −1790.39 | |
| F | 0.12 | | 13.01 | | 8.51 | | 9.99 | | 9.6 | |
| (DF(m), DF(r)) | (1, 1016) | | (9, 1008) | | (16, 1001) | | (12, 1005) | | (13, 1004) | |
| Prob > F | 0.73 | | **<0.001** | | **<0.001** | | **<0.001** | | **<0.001** | |

N = 1017; CI = confidence interval; * $p < 0.05$, ** $p < 0.01$, *** $p < 0.001$; AIC = Akaike information criterion; AIC/n = Akaike information criterion adjusted for sample size; BIC = Bayes information criterion. Preferred model fit statistic in bold.

Our preferred model for explaining changes in anger towards Muslims is therefore Model 2, using a combination of pseudo-R squared values and information criteria, and preferring models with less explanatory variables where differences between model fit statistics were minor (comparison of the BIC statistics in particular provide very strong support for preferring Model 2 over all other models). While the number of hours of news consumed is not statistically significant in Model 2, religiosity, politics, knowledge about Muslims, and personal engagement with Muslims all have a statistical impact on anger towards Muslims in the expected directions. However, all effects are small. Only political viewpoint (including its interaction with religiosity) has a notable effect on anger towards Muslims, where a move from the extreme left to the extreme right of politics predicts a movement of three steps of increased animosity along the underlying latent variable of anger (a marginal change of 3.05 ($p < 0.001$)) while holding all other explanatory variables at their average.

To demonstrate the effect of the interaction between religiosity and political viewpoint on animosity towards Muslims, predictions of anger towards Muslims are calculated using the user-written *mtable* routine in Stata 16.1 after fitting Model 2 (Long and Freese 2014), and are depicted first in Figure 1. When the participant does not identify adherence to a religion, as shown in the left hand panel, a move from the extreme left to the extreme right of political viewpoint moves the predicted level of anger on the scale almost five steps of increased animosity on the underlying latent variable (a marginal change of 4.78 ($p < 0.001$)). However, that relationship is attenuated where religion is moderately important to the participant (in the center panel), and disappears where religion is of high importance (in the right hand panel).

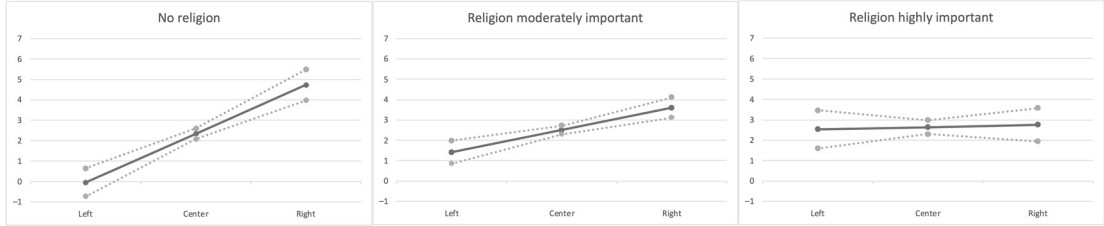

**Figure 1.** Predicted anger towards Muslims (estimate and confidence interval from Model 2) according to religiosity and political viewpoint: no religion to high importance. Note: predictions are for the latent underlying anger variable, and therefore allow for predictions outside the bounds of 1 to 7.

In Figure 2, the same relationship is depicted using the predicted change in anger according to religiosity for participants who placed themselves at the extreme left, center, and extreme right of political viewpoints. In the center panel, for those in the center of politics, anger does not change significantly between levels of religiosity. For those on the left of politics, a move from no religious affiliation to high religiosity predicts a 2.5 level increase in anger towards Muslims (a marginal change of 2.57 ($p < 0.001$)), while on the right, the same change in religiosity predicts an almost two level *decrease* in anger (a marginal change of −1.97 ($p = 0.002$)). As such, it is clear that our decision to examine the confirmation bias hypothesis through how the effect of news consumption on anger towards Muslims changed over both political viewpoint and religiosity was appropriate.

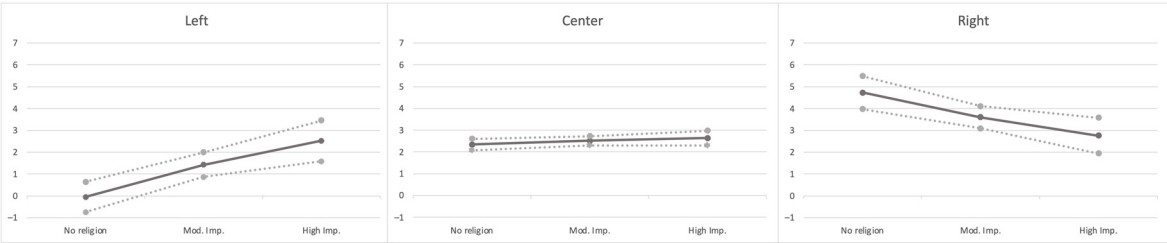

**Figure 2.** Predicted anger towards Muslims (estimate and confidence interval from Model 2) according to religiosity and political viewpoint: left to right political views. Note: predictions are for the latent underlying anger variable, and therefore allow for predictions outside the bounds of 1 to 7.

## 8. Discussion

At the outset of this paper, we were curious about whether the amount of news non-Muslim Australians consumed affected their levels of anger towards Muslims. While there is a significant body of research that highlights the negative nature of much news media coverage of Muslims, and the various contributing factors to anti-Muslim sentiment, there are very few studies that we could locate that examine whether the amount of news media people consume influences their attitudes towards Muslims. This article was prompted by one of the few studies to examine the issue of the effect of news consumption levels on attitudes towards Muslims, and so our discussion is informed by that study. That study by Shaver et al. (2017) was undertaken in New Zealand, a country with a similarly small percentage of Muslims as Australia. The findings of Shaver and colleagues' study suggest that there is a relationship between the amount of news New Zealanders consume and the levels of anger they feel towards Muslims. While the evidence for the association in Shaver et al.'s study is statistically significant, the substantive effect on anger of news consumption was small (with anger towards Muslims measured on a Likert scale spanning 1 to 7, a change from 0 to 15.49 h of news consumption per week increased the expected anger from a score of 3.10 to 3.21). The authors also found that relationship to be consistent across the political spectrum, leading the authors to conclude that confirmatory bias in their participants' choices of news consumption was not driving anti-Muslim sentiment.

That research prompted us to explore whether the amount of news survey participants consumed was a significant factor in the extent of anger held by non-Muslim Australians towards Muslims, while controlling for demographic factors identified through the extant literature that might play a role in anger levels. We also looked at political orientation and religiosity as possible factors that may attenuate the effects of news consumption on anger, as well as other possibly important drivers of anti-Muslim sentiment identified in the literature.

Unlike Shaver and colleagues, we did not find a relationship between amount of news consumption and attitudes towards Muslims. Specifically, we did not find that those in our survey who consumed higher self-reported amounts of news had higher levels of anger towards Muslims. This may be because there are cultural differences between New Zealanders and Australians, or because of an as-yet unidentified difference in the news reporting available in the two countries. Interaction with political viewpoint and importance of religion did not affect the lack of relationship, nor did interaction with time spent with Muslim people. It is useful to reflect on Arlt's (2021) findings in the context of what we discovered here, with Arlt identifying in her study that the political leanings of her study participants had a significant role in their intention to support a vote to place a ban on Muslim women veiling. However, study participants' exposure to political information via news media did not influence their voting intentions.

While we did not find evidence to support the existence of either media effects or confirmation bias in our nationally representative Australian sample when examining animosity towards Muslims, the operationalization of these concepts may be responsible. Without knowing how much news media *of what type* is consumed by participants, the

overall number of hours spent consuming news media may not allow for differentiation of the mechanisms at work. However, Shaver et al. (2017) did find evidence to support both theories using the same operationalizations of both news media consumption and animosity towards Muslims, a relationship we would expect to hold in our data; and given the historically negative nature of reporting on Islam in Australia and the lack of evidence that the relationship between hours of news consumed and anger towards Muslims changes across the political spectrum or religiosity, such a distinction between news sources with differing ideological media framing may be less relevant than we might assume.

While our modelling strategy differed from Shaver et al. (2017) due to both unavailability of the measures of attitudes towards other outgroups and size of sample, we did also obtain extra measures of variables previously shown to affect anti-Muslim attitudes in time spent with, and knowledge about, Muslim people, increasing our ability to isolate the effect of news consumption. We also recognized the prominent interaction between political viewpoint and religiosity in attitudes present in Australian populations and controlled for that element when examining possible confirmatory biases. In addition, our null finding for the influence of news consumption on anger towards Muslims was consistent across models, leading us to expect that missing variable bias is not likely to be affecting our theoretical findings. If media effects or confirmatory bias are in fact important predictors of animosity towards Muslims, those relationships require much more nuanced measures.

For those theorizing the relationship between conflict expressed as anger towards Muslims, the roles of political persuasion and religiosity, and the interconnections with news consumption, there are lessons here. Key amongst them is that increased news consumption has, at least for our Australian study participants, no relationship with increased anger towards Muslims. For those approaching research into attitudes towards Muslims and news consumption using media effects theory, our findings suggest that at least in Australia and in the case of attitudes of non-Muslims towards Muslims, amount of news consumption is not a factor, and researchers approaching such issues from a media effects perspective should look at aspects of engagement with news other than levels of consumption. Researchers using media effects theory to explore questions about the influence of news media consumption in Western contexts on non-Muslims' attitudes towards Muslims, including levels of anger or warmth and acceptance or rejection of the presence of Muslims, should consider whether focusing on the amount of news consumed is enough to reveal the nuances of media effects. A range of other factors may be at work, and researchers may need to focus on those including the nature of the news media being consumed, i.e., conservative or more liberal, the type of news stories, for instance, hard news, feature articles, human interest or soft news, whether news consumption is designed to reinforce or challenge non-Muslims' beliefs and attitudes, and whether engagement with social media rather than news media has more influence on attitudes.

There is a wealth of research that highlights that religion can be a flashpoint in conflicts and this is particularly the case in many Western democracies where the presence of Muslims caused tensions. Coupled with this, some politicians used these tensions to create further division between Muslims and non-Muslims for political gain. Our study showed there are notable effects in relation to the connections between political persuasion and decreased anger (and potentially conflict) amongst those who are on the left to extreme left of politics and increased anger amongst those who are on the right to extreme right of politics. When it came to religion, amongst the participants who said religion was of high importance to them, this had a levelling effect in extent of anger across the political spectrum. As such, religiosity appears to temper outgroup anger on the right of politics, and increase outgroup anger on the left. Our recommendation therefore is that researchers consider the interaction between political ideology and religiosity in multicultural Western democracies like Australia, where in fact the increase in secularization may indicate greater influence of political viewpoint (rather than religion) on attitudes towards outgroups, particularly religious minorities. Care must be taken to avoid conflating religiosity with

ideology, or assuming a constant effect of each individually, particularly outside the United States of America.

## 9. Conclusions

As with any study, ours has its limitations. We set out to investigate whether amounts of news consumed affected non-Muslims' attitudes towards Muslims. We did so because many studies indicated there are connections between non-Muslims' news media consumption and attitudes towards Muslims, and the problematic nature of reporting on Muslims was established so we might expect that consuming more of that reportage would result in less favorability towards Muslims. We posited that, given the findings of existing research, we would find some connections between news media consumption and attitudes towards Muslims. However, we did not. Like previous studies investigating support for media effects theory in this area, we did not ask study participants what kind of news they consumed, for example tabloid or quality news media. It may be that the lack of effect of quantity of news consumption on anger provides support for the uses and gratifications theory (Blumler and McQuail 1969), that is, we choose to consume types of news media that support the views we already have and create our own feedback loop.

It may be the case that non-Muslims in Australia consume news that reinforces their attitudes towards Muslims and in the case of those who already hold levels of anger towards Muslims the amount of news they consume has no effect on their attitudes. Alternatively or in addition, views towards Muslims may be shaped more by the continual interaction with media that occurs throughout daily life than by consumption of traditional news media. Future examination of how media frames attitudes towards religious and cultural outgroups like Muslims should therefore attempt more nuanced measures than the amount of news media consumed in order to examine media effects theory's mechanisms.

To this end, future studies should focus on the type of news consumed and whether that influences levels of anger or warmth non-Muslims hold towards Muslims, and whether different styles and deliveries of news might play a role in prompting positive changes in attitudes towards Muslims. When it comes to the role of journalism and religious convictions in fueling conflict, our study reinforces the role of religiosity in attitudes towards Muslims and highlights that when religion and politics are considered together in a secularizing Western democracy, religion has a differing effect on anger towards Muslims across the spectrum of political allegiances (far right to far left). Importantly for those interested in how journalism contributes to religious conflict, our study highlights that the amount of news consumed does not have a role in increasing anger towards Muslims.

**Author Contributions:** Conceptualization, J.E. and S.W.; methodology, S.W.; formal analysis, S.W.; writing—original draft preparation, J.E. and S.W.; writing—review and editing, J.E. and S.W.; visualization, S.W.; project administration, J.E.; funding acquisition, J.E. All authors have read and agreed to the published version of the manuscript.

**Funding:** This research was funded by the Commonwealth of Australia. No research grant number.

**Institutional Review Board Statement:** This study was approved through Griffith University's Human Research Ethics Review process, protocol 2014/531.

**Informed Consent Statement:** Informed consent was obtained from all subjects involved in the study.

**Data Availability Statement:** The data is held by the Chief Investigator Jacqui Ewart.

**Conflicts of Interest:** The authors declare no conflict of interest.

## Appendix A. Survey Questions: General Knowledge about Muslims

Which of the following comes closer to your view. Would you say that, compared with other religions, the Islamic religion . . . ?

1. Is more likely than others to encourage violence
2. Does not encourage violence more than others

3.	(Neither/other view)/Neither/other view
    (Do not know)/Not sure
    (Refused)/Prefer not to say

How much do you know about the religion of Islam?

1.	A great deal
2.	Some
3.	Not very much
4.	Nothing at all
    (Do not know)/Not sure
    (Refused)/Prefer not to say

Which of the following comes closer to your view ... ?

1.	Muslims should have same rights as other groups to build houses of worship in local communities
2.	Local communities should be able to prohibit construction of mosques if they do not want them
    (Do not know)/Not sure
    (Refused)/Prefer not to say

How many Muslim people do you know?
*If you're unsure of the exact number, please give your best estimate.*

1.	Response given (RANGE: 1–1000)
2.	None
3.	(Unsure of number but know at least one Muslim person)
    (Do not know)/Not sure
    (Refused)/Prefer not to say

How often are you in the company of one or more Muslim people?

1.	Daily
2.	Weekly
3.	Monthly
4.	Less than monthly
5.	Never
    (Do not know)/Not sure
    (Refused)/Prefer not to say

To the best of your knowledge, which of the following are associated with Islam?
*Please select all that apply*

1.	Sufi (PRONOUNCED: Soo-fee)
2.	Sunni (PRONOUNCED: Soo-nee)
3.	Sikh (PRONOUNCED: Seek)
4.	Shi'ite (PRONOUNCED: She-ite)
5.	(None of the above)
    (Do not know)/Not sure
    (Refused)/Prefer not to say

To the best of your knowledge, which of the following statements correctly describes the meaning of Sharia Law?
*Please select all that apply*

1.	Islamic law that permits Muslims to punish non-Muslims or infidels
2.	Guidelines that cover religious, ethical, moral, spiritual, legal, economic and political aspects of a Muslim's life
3.	A divine law followed by Muslims that overrides laws made by parliament and courts
4.	(None of the above)
    (Do not know)/Not sure
    (Refused)/Prefer not to say

To the best of your knowledge, which of the following religious figures does Islam recognize as prophets?
*Please select all that apply*

1. Adam
2. Abraham
3. Moses
4. Jesus Christ
5. Muhammad
6. (None of the above)
       (Do not know)/Not sure
       (Refused)/Prefer not to say

To the best of your knowledge, is Islam similar in many ways to Judaism and Christianity?

1. Yes
2. No
       (Do not know)/Not sure
       (Refused)/Prefer not to say

To the best of your knowledge, which of the following actions might be offensive to some Muslims or seen as inappropriate behaviour?
*Please select all that apply*

1. Publishing an image of the Prophet Muhammad (PBUH)
2. Offering only pork at a function where Muslim people are invited
3. Wearing shorts or a short skirt inside a mosque
4. Extending your hand to shake hands with a member of the opposite gender
5. Making jokes about Allah (God) and prophets
6. (None of the above)
       (Do not know)/Not sure
       (Refused)/Prefer not to say

To the best of your knowledge, which of the following statements about Muslims in Australia are correct?
*Please select all that apply*

1. All Muslim women in Australia wear a veil or head covering
2. Muslims were in Australia before European settlement
3. Muslims are one of the most ethnically diverse religious groups in Australia
4. Muslims make up less than 5 percent of the Australian population
5. Muslims are routinely negatively stereotyped by the mainstream news media
6. (None of these are correct)
       (Do not know)/Not sure
       (Refused)/Prefer not to say

To the best of your knowledge which of the following terms mean something that is permitted for Muslims?
*Please select all that apply*

1. Kosher (PRONOUNCED: Koh-sher)
2. Haram (PRONOUNCED: Hah-ram)
3. Halal (PRONOUNCED: Hah-lal)
4. Salaam (PRONOUNCED: Sah-larm)
5. (None of the above)
       (Do not know)/Not sure
       (Refused)/Prefer not to say

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
