# Peer review of "Exploring the Influence of News Consumption on Non-Muslim Australians’ Attitudes towards Muslims"

_religions, doi:10.3390/rel13080744_

Round 1

Reviewer 1 Report

Dear authors, 

I really enjoyed reading your manuscript. The topic is very pertinent and interesting. The manuscript has a very solid and well-structured literature review. However, I believe that the manuscript needs several changes.

1- I start with the abstract. It is important to improve the abstract. The objectives are not clear and it should focus on the main findings of the manuscript.

2- Regarding the literature, there is a lot of information. This information would be better organized if hypotheses were presented throughout the text. The hypotheses must be presented after each theoretical foundation.

3- Regarding the methods, the analysis plan contains the research questions. I think it would be more appropriate to put the research questions in the introduction to the manuscript. I also suggest that the authors create a table containing all the variables. In the "measurements" section, it would be more understandable for the reader to be able to have a table with the variables and respective scales, averages, etc...

The authors also report that "A series of questions designed to test basic knowledge about Muslims was asked, enabling participants to obtain a score out of 14. On average, participants scored 10.22 out of 14 (SD=2.35)". What are these questions? They must be in the manuscript.

4- In the results it is not very clear what is the influence of political orientation on the attitude towards Muslims. Or what is the relationship of sociodemographic factors? Is the attitude different between genders? Among older and younger people? Between the most educated and the least educated?

5 - The discussion must be improved. Throughout the discussion, the authors only refer to one work [Slaver et al. 2017].

Author Response

Thank you for your helpful comments. We have amended our manuscript as per your following suggestions:

Added objectives of the study to the abstract as per Rev 1.

Placed the research questions into the introduction of the paper as per Rev 1 suggestion.

As per Reviewer 1’s request included the questions asked in oour survey about general knowledge of Muslims and their faith. We have placed these at the end of the references.

We have added to the discussion as per Reviewer 1’s comment about the discussion relying on the Shaver et al study, we have clarified in the discussion section that this is because the Shaver study is the only other study to explore levels of news consumption and their effect on attitudes towards Muslims. We have pointed out that this is why we focus on that article in the discussion section

Reviewer 2 Report

The authors address a theme that continues to be relevant in the social sciences (sociology, social psychology, anthropology, communication) in particular in the study of the alleged role of the media in influencing negative attitudes towards Muslims or Islam. The precise critical review of the literature on the subject that the authors provide in an extensive way (six dense pages of the nineteen overall of the paper) confirms that by recalling how some research results are now acquired and, therefore, it does not make much sense to question them. The article by Shaver et al. (2017, in PLos 12/3), in particular, to which the authors refer to argue on the working hypothesis they intend to falsify, moves, in fact, from the finding widely accepted by scholars about the fact that people tend to interpret news they receive from the media on the basis of pre-existing prejudices that oriented their negative attitudes towards this or that stigmatized group. This article talks about the New Zealand situation. Here the residents of immigrant origin of Muslim faith are relatively few, 1.2% of the total population. The serious attack on the al-Nour mosque in the city of Christchurch, carried out in March 2019 by a member of a far-right group of nationalist supremacist tendencies, raises a question that the authors of the article should have asked themselves: what is the role of social media in facilitating forms of political aggregation and mobilization that target immigrants, particularly those who are identified as Muslims. In other words, instead of analysing the connection between news consumption and aggressive feelings (of anger, as the authors specify, as if they wanted to take up the well-known title of a  pamphlet by Oriana Fallaci - The Rage and the Pride (2002)- written straight away from this well-known Italian journalist in the aftermath of 9/11, it would have been preferable to focusing on the ideology of the far-right movements (which circulate furiously between the two sides of the Atlantic) that paint the face of the enemy of white-western civilization -Christian by superimposing the two faces of the foreign immigrant with that of the faithful and zealous Muslim. The arguments aimed at falsifying the working hypothesis would have been more convincing. The authors themselves recognize it with great objectivity in the discussion of the data when they say on p. 14 "without knowing how much news media of what type is consumed by participants...". After all, the most consistent predictor of animosity attitudes towards Muslims  - Islamophobia is a more complex notion that the authors rightly hold in the background preferring to talk about what non-Muslims think of Muslims in a country like the Australia where this latter constitutes, as in New Zealand, a modest and not particularly visible presence - is measured on the basis of the political self-positioning of the interviewees. Therefore, it would be interesting if the authors added some more information about the presence of white supremacist movements and whether they are connected via transnational social media or not. 

In conclusion, the article is a valid contribution to research on the formation of anti-Muslim attitudes, rigorously organized between the theoretical part, presentation of the methodological choices, and, finally, discussion of the results. However, it would gain clarity if the authors clarify the two aspects mentioned above, I sum up here:

- when we talk about news consumption it would be appropriate to introduce the distinction between old and new media, in particular, social media which seem to have a more relevant role in the production and circulation of news aimed at arousing "hatred" and "contempt" towards of ethnic or cultural minorities;

- when it is highlighted in the discussion of the results of the statistical weight in the interpretation of the empirical data collected with the survey of the ideological-political factor compared to the other hypothesized indicators, it would be appropriate to provide the reader with some more information on the consistency of extreme right-wing supremacist movements in Australia (as well as in New Zealand, perhaps inserting a note, in particular, recalling the attack that took place in 2019, two years after the publication of the article by Shaver et al. In 2017).

Author Response

We thank you for your comments about our article. We have added a sentence to the introduction clarifying that we are NOT studying the role of social media in anger towards Muslims. We feel most of the comments were about white supremacy which is not the focus of our paper. It is not possible to determine whether a person who indicates they are to the right of the political spectrum is a white supremacist. We felt that this would significantly change the focus of our article but we believe there may be something further for us to explore in relation to white supremacy in future studies and surveys.

Reviewer 3 Report

This article addresses a highly important area and adds valuable results by qualifying the hypothesis about correlations between amount of news consumption and access to other means of information such as interaction in daily life with Muslims, and negative attitudes, anger, towards Muslims. The article furthermore present interesting new findings regarding the connection between political views and religiosity, and the need in further research to analyze what type of news media that is consumed with regard to negative attitudes to Muslims.

Some comments for revision that would improve the clarity and merits of the article:

- The balance between previous research and the study that is presented. The review of previous research now encompasses half of the article. It is very comprehensive, however it is not clear if and how all of the studies mentioned are relevant for the actual study at hand. There are also a number of unnecessary repetitions when moving between sections/studies. My recommendation is to condense and shorten this part, and end with a summary of what results that inform the research questions of the presented study.

- Theory; it is not entirely clear until in the methods section whether the authors aim to use or test/evaluate media effects theory. How this theory  is used can be clarified in the introduction to the article, I e what "elements " and hypotheses from previous research using media effects theory will be tested? Following from this comment, the research questions, as they are now written, needs to be clarified with regard to what hypotheses on connections between variables that they are meant to test. One example is the question “If confirmatory bias exists, political viewpoint should neutralise media effects on anger (and political viewpoint is inextricable from religiosity when examining attitudes towards Islam and Muslims in Australia)”. What confirmatory bias means, I e consuming news strengthen already held political or religious views/experiences, is explained on page 12-13 but should be made clear in the research questions. Also, the sentence in brackets blurs the clarity of what correlations that are to be analyzed in the article.

Author Response

In response to your comments which were very helpful, we have cut down the literature review, removed repetition and redundant sentences from segues and added a short par at the end of the literature section indicating relevance of specific literature to our study. We have also added that we are using media effects theory in the introduction. We have adjusted one of our hypotheses in response to your comment. 

Round 2

Reviewer 1 Report

The reviewers added my suggestions to the manuscript. The article has improved and can be published.